# Identification of Apple Orchard Planting Year Based on Spatiotemporally Fused Satellite Images and Clustering Analysis of Foliage Phenophase

**Yaohui Zhu** [1,2,3] **, Guijun Yang** [1,2,]***, Hao Yang** [1,2]**, Jintao Wu** [1,2] **, Lei Lei** [1,2]**, Fa Zhao** [1,2]**, Lingling Fan** [1,2] **and Chunjiang Zhao** [1,2,3]

1 Key Laboratory of Quantitative Remote Sensing in Agriculture of Ministry of Agriculture, Beijing Research Center for Information Technology in Agriculture, Beijing 100097, China; yaohui_zhu@bjfu.edu.cn (Y.Z.); yangh@nercita.org.cn (H.Y.); e16101001@stu.ahu.edu.cn (J.W.); 17210064045@stu.xust.edu.cn (L.L.); p18101005@stu.ahu.edu.cn (F.Z.); p17301156@stu.ahu.edu.cn (L.F.); zhaocj@nercita.org.cn (C.Z.)
2 National Engineering Research Center for Information Technology in Agriculture, Beijing 100097, China
3 School of Information Science and Technology, Beijing Forestry University, Beijing 100083, China
* Correspondence: yanggj@nercita.org.cn

**Abstract:** The planting year of apple orchard not only determines the fruit output but also provides information for the governmental management of the fruit industry. However, considering that different orchards use different management and cultivation methods, this may result in some trees having similar outlines but different planting years, and it is, therefore, difficult to effectively determine the actual planting year based on textural or structural characteristics. Therefore, the monitoring method provided in this paper is not to monitor the growing year positively from the planting of orchard seedlings but to use time series remote sensing data to reverse determine the continuous growth age of each existing orchard. The city of Qixia, Shandong Province, China, was used as a case study. Firstly, the spatial distribution of apple orchards was accurately extracted using the Sentinel-2 normalized difference vegetation index (NDVI) spatiotemporally fused images and phenological vegetation information. Secondly, using region of interest (ROI) data for different vegetation types obtained from a field survey, NDVI time series were extracted from the Sentinel-2 NDVI spatiotemporally fused image. Among them, three characteristic phenological periods were selected, and the NDVI time series for apple orchards was used as a template to extract the apple orchard distribution area from 2000 to 2017. Then, the distribution area of apple orchards was defined as the area of interest in the planting year, combined with the Landsat NDVI time series image composed of three characteristic phenological periods each year from 2000 to 2017, and the apple orchard phenological curve. Subsequently, a Euclidean distance (ED) method was used to calculate the distribution area of apple orchards for each year between 2000 and 2017. Finally, a pixel-by-pixel inverse time series calculation method was used to obtain the planting year of apple orchards in the study area. This study provides a new way to accurately identify the planting year of apple orchards using satellite remote sensing images.

**Keywords:** apple orchard; planting year; remote sensing; spatiotemporal fusion; phenology

---

## 1. Introduction

Apple orchards are one of the most important agricultural production types in the world. Especially, China has a planting area and an annual output of 2.22 million hectares and 41.39 million tons, respectively, each of which accounts for more than 40% of the world total [1]. In China, the aging of apple orchards is one of the most important factors affecting the yield and quality of apple

crops, and can directly or indirectly reduce the economic output of orchards. Therefore, the accurate determination of apple planting age is very important for governmental agencies to predict regional apple production and to assist in the formulation of reform policies for aging orchards. Traditionally, the acquisition of planting age in apple orchards requires the collection of farm planting records or destructive sampling methods. However, although this approach can ensure accurate estimation, it is time-consuming, labor-intensive, and difficult to apply on a large scale [2,3].

With the development of remote sensing technology, remote sensing data with different temporal and spatial resolutions provide the possibility to classify vegetation and determine its spatial distribution [2,4–7]. Remote sensing data have been widely used to determine the age of various types of vegetation. Three main approaches are used to extract vegetation age from remote sensing data: (i) approaches using remote sensing images. This approach assumes that there are significant differences in spectral characteristics for different vegetation ages—for example, spectral indexes or texture information [8–13]. Additionally, WorldView-2 satellite images have been used to estimate the area and age of oil palm canopy using a regression model [14]; (ii) approaches using lidar point cloud data. In forest ecosystems, information of vertical forest structure is often used to determine vegetation age [15]. Racine et al. [16] used airborne lidar data to extract structural information from forests in Quebec, Canada, and thereby estimate forest age using the k-nearest neighbor method. Iizuka et al. [17] used remote sensing data from the Advanced Land Observation Satellite (ALOS) phased array L-band synthetic aperture radar (PALSAR) to extract structural information for three forest types and thereby estimate forest age; (iii) approaches which combine remote sensing images and lidar point cloud data. For example, Rizeei et al. [18] proposed a method for estimating the age of oil palm forests using a combination of Worldview-3 satellite imagery and airborne lidar data. The four kernel functions of a support vector machine (SVM) model were compared to obtain the best contour of tree canopy coverage, and a statistical regression model was constructed by combining canopy and structural parameters to accurately estimate the forest age. Furthermore, Tan et al. [19] used images from the United Kingdom-Disaster Monitoring Constellation-2 (UK-DMC-2) and ALOS PALSAR satellites to extract the age of oil palm forests in Southern Malaysia. The results indicate that height data may be important variables for estimating the age of oil palm trees.

Most previous research on the identification of forest age using remote sensing data has focused on spectral characteristics, textural characteristics, or forest structure parameters during the vigorous growth period of trees. However, this approach requires remote sensing images with a high spatial resolution and high-density lidar point clouds, which limits the size of the monitoring area. Additionally, since different orchards may have different management and cultivation methods, some fruit trees may have similar outlines but different ages, which makes it difficult to judge the actual planting age based on spectral, textural, or structural characteristics. Therefore, there is an urgent need for an effective method for identifying the planting year of apple orchards.

The prerequisite for the identification of the planting year of apple orchards is to accurately determine the spatial distribution of the orchards. Considering the potential complexity of land cover and the spectral similarity between different species of vegetation, using only a single image to classify regional vegetation types can be severely restrictive [20,21]. In recent years, a large number of studies have attempted to classify land cover using time series of remote sensing images [22–26]. Compared with traditional classification methods, this approach more comprehensively considers the phenological characteristics of plants and offers improved classification accuracy. However, remote sensing images are often disturbed by cloud cover, which makes it difficult to obtain high-quality satellite time series data with a medium or high spatial resolution [27]. Therefore, Gao et al. [28] proposed the spatial and temporal adaptive reflectance fusion model (STARFM) algorithm, which combines Moderate Resolution Imaging Spectroradiometer (MODIS) data with a high temporal resolution and Landsat data with a high spatial resolution to generate a fused dataset with a high spatial and temporal resolution. Many studies have applied this algorithm for land cover classification [27,29,30]. However, this method has some limitations for heterogeneous landscapes [31,32]. Subsequently, Zhu et al. [33] proposed the

enhanced STARFM (ESTARFM) fusion algorithm, which can more accurately generate fused images with a high temporal and spatial resolution; compared with the original STARFM algorithm, this algorithm can retain more spatial details, especially for heterogeneous landscapes.

This paper proposes a new method for the identification of the planting year of apple orchards on a regional scale. This method is implemented in two steps: (i) extract the spatial distribution of apple orchards. In this step, the ESTARFM algorithm is used to fuse simulated 2018 Sentinel-2, and MODIS normalized difference vegetation index (NDVI) images to generate NDVI time-series data. By combining the fused image with phenological vegetation features generated using the TIMESAT 3.3 toolbox, an SVM supervised classification method was used to extract the spatial distribution of apple orchards; (ii) identification of apple orchard planting year. Using region of interest (ROI) data for different vegetation types (apple orchard, cherry orchard, vineyard, coniferous forest, and broad-leaved forest) obtained from a field survey, the NDVI time series were extracted from the Sentinel-2 NDVI spatiotemporal fused image. Among these NDVI time-series curves, three phenological characteristic periods that can be used to determine the distribution of apple orchards were selected, and the NDVI time series of the apple orchard was used as a template for extracting the apple orchard distribution. Then, the spatial distribution of apple orchards in 2018 was defined as the area of interest in the planting year, and Landsat NDVI time-series images composed of three phenological characteristic periods from 2000 to 2017 were masked, respectively. Subsequently, according to the apple orchard template, the distribution area of apple orchards from 2000 to 2017 was calculated using the Euclidean distance (ED) method. Finally, the pixel-by-pixel inverse time series calculation program developed in the MATLAB software was used to obtain the planting year distribution of apple orchards in 2018.

## 2. Materials and Methods

### 2.1. Study Area

The city of Qixia, Shandong Province, China, was selected as the study area. This city is one of the main apple-producing areas in China. Qixia is located between 37°05′–37°33′ N and 120°33′–121°15′ E and covers an area of approximately 2016 km$^2$ (Figure 1). This area is characterized by a temperate continental monsoon climate with a total annual rainfall of 650 mm and an annual mean temperature of ~11.3 °C. The city is dominated by hills and has an average elevation of 178 m above sea level. The annual cumulative sunshine hours are 2690 h. The temperature difference between day and night is large in the autumn, and the soil and environmental conditions are very suitable for apple growth. In addition to apples, the area also contains cherry, grape, pear, and peach cultivation, as well as broad-leaved and coniferous forests.

### 2.2. Field Data

During the growing season, the main phenological stages which affect the bulk spectral behavior of apple orchards are (1) dormancy period (DOY 49–97): the date at which photosynthetic activity and green leaf area begin to rapidly decrease; (2) growth period (DOY 129–193): the date when photosynthetic activity begins; and (3) peak period (DOY 225–305): the date at which the plant green leaf area is maximum. To obtain training and validation samples for the land cover classification in the study area, as well as the information on the planting year of apple orchard, field data collection was carried out during the growing period from August to September 2019. The geographical coordinates and the corresponding land cover types of the study sites were collected using a Qianxun position SR2 satellite-based RTK receiver mobile device with a centimeter-level positioning accuracy (Qianxun Spatial Intelligence Inc., Zhejiang, China). Additionally, high-resolution Google Earth images acquired in 2019 were used to supplement the sample size of different types of regional surveys. These Google Earth images were used to select the actual distribution areas of different land cover types by visual interpretation (Figure 2). We surveyed 280 field sites in a relatively homogeneous area with uniform land cover around the sites. This area contains a range of land cover and vegetation types, including

apple orchards, cherry orchards, vineyards, broad-leaved forests, coniferous forests, water, and urban areas. A total of 40 field sites were investigated for each of these seven land cover types (Figure 1).

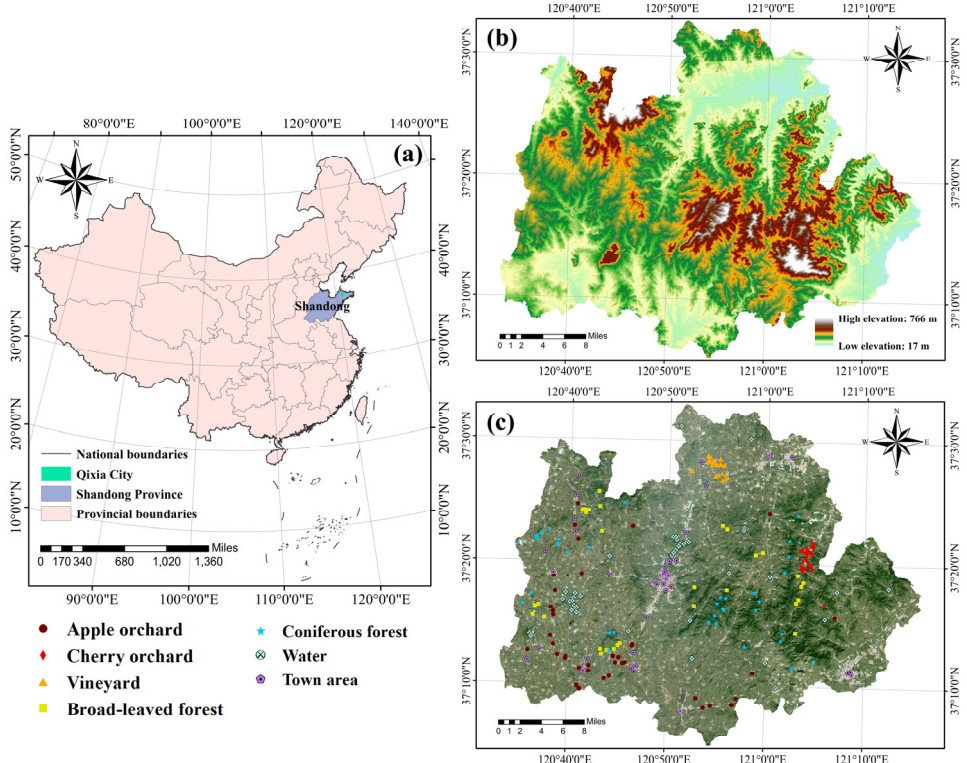

**Figure 1.** (**a**) Location of the study area; (**b**) Topography of the study area; (**c**) Location of the samples on the field.

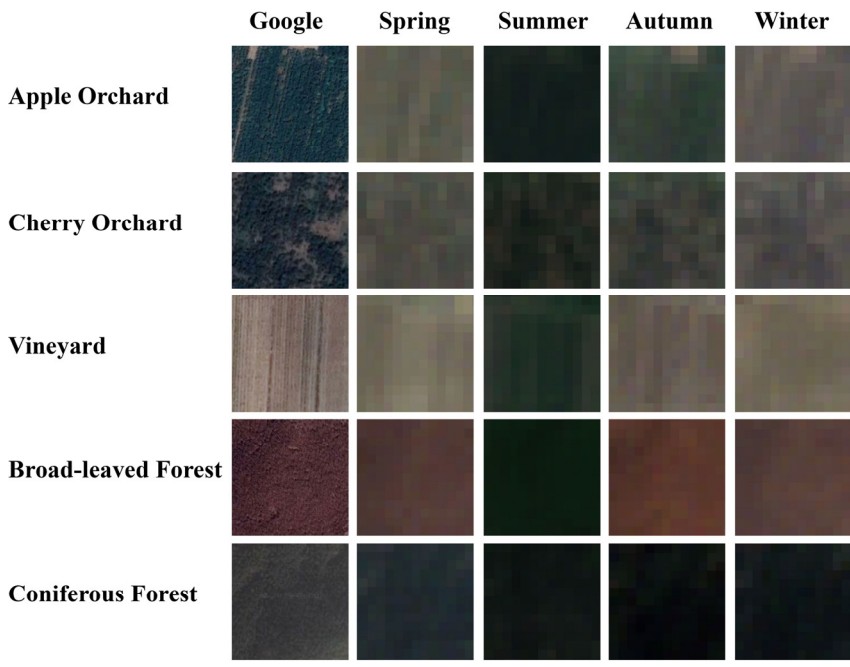

**Figure 2.** Seasonal differences among major vegetation types in the study area. The first column of images shows Google Earth images; the second to fifth columns show the Sentinel-2 images of spring, summer, autumn, and winter, respectively.

## 2.3. Remote Sensing Data Acquisition and Preprocessing

The remote sensing images used in this study were Sentinel-2, Landsat TM/ETM+/OLI, and MODIS Terra Vegetation Index (MOD13Q1) products. The data set list of Sentinel-2 and Landsat is shown in Figure 3. In this paper, we name the satellite image data by measuring the number of days between the shooting time of the satellite image and the 1st of January of the year.

| Year | Dormancy period | | | | 113 | Growth period | | | | | 209 | Peak period | | | | | |
|---|---|---|---|---|---|---|---|---|---|---|---|---|---|---|---|---|---|
| | 49 | 65 | 81 | 97 | 113 | 129 | 145 | 161 | 177 | 193 | 209 | 225 | 241 | 257 | 273 | 289 | 305 |
| 2000 | | 68-7 | | | | | | 164-7 | | | | | | 260-7 | | | |
| 2001 | | | 78-5 | | | | 142-7 | | | | | | | | 270-5 | | |
| 2002 | | 65-5 | | | | | 145-5 | | | | | | | | | 289-5 | |
| 2003 | | | 76-7 | | | | | | | | | | | | | 292-5 | |
| 2004 | | | | 95-7 | | 127-7 | | | | | | | | | 279-5 | | |
| 2005 | | 65-7 | | | | | | | | | | | | 265-5 | | | |
| 2006 | | | 84-7 | | | | | | | | | | | | 276-7 | | |
| 2007 | | | | | 119-7 | | | | | | | | | 255-5 | | | |
| 2008 | | 66-5 | | | | 122-7 | | | | | | | | 258-5 | | | |
| 2009 | | | 76-7 | | | | | | | | | 228-5 | | | | | |
| 2010 | | | 87-5 | | | 127-7 | | | | | | | | | 279-5 | | |
| 2011 | | 66-7 | | | | | | 162-7 | | | | | | | 266-5 | | |
| 2012 | | | 85-7 | | | | 149-7 | | | | | | | | 277-7 | | |
| 2013 | | | 79-8 | | | | 143-8 | | | | | 231-7 | | | | | |
| 2014 | | 66-8 | | | | | 146-8 | | | | | | | | | 290-8 | |
| 2015 | | 69-8 | | | | | 133-8 | | | | | | | | 269-7 | | |
| 2016 | | 72-8 | | | | | | 168-8 | | | | | | | 280-8 | | |
| 2017L | 58-8 | | | | | | 138-8 | | | | | | | | 266-8 | | |
| 2017S | | | | 91 | | 131 | | | | | | | | | | 296 | |
| 2018 | | 72 | 92 | 99 | 117 | | 139 | 157 | 172 | | | | | 252 | | 292 | 302 |

**Day of year**

**Figure 3.** The long-term Landsat image dataset (2000–2017) and Sentinel-2 image dataset (2018) used in this study. In the Landsat dataset, the suffixes −5, −7, and −8 represent the Landsat thematic mapper (TM), enhanced thematic mapper plus (ETM+), and operational land imager (OLI) sensors, respectively. Among them, 2017L and 2017S represent the data of Landsat and Sentinel-2 in 2017, respectively.

The blue, green, red, and near-infrared bands of Sentinel-2 images have a spatial resolution of 10 m and a 5–10-day return period. The images are freely available from the European Space Agency (ESA) Copernicus Open Access Hub (https://scihub.copernicus.eu/without/dhus/#/home). These images have been widely used in land cover detection [34,35], natural disaster monitoring [36], forest monitoring [37], and agricultural monitoring and management [38,39]. The Sentinel-2 Atmospheric Reflective Top (L1C) product was used in this study. This product is an atmospheric apparent reflectance product which has been ortho-corrected and geometrically fine-corrected but has not yet been atmospheric-corrected. The Sen2Cor-2.5.5 plug-in was used to convert the Sentinel-2 L1C product to the corresponding Atmospheric Corrected Bottom (L2A) product to eliminate the atmospheric effects. Then, the L2A product was resampled in the Sentinel application platform (SNAP 6.0.10), and the blue, green, red, and near-infrared bands were converted to ENVI format. The NDVI was calculated using the surface reflectance of the Sentinel-2 satellite images according to Equation (1) [40].

The Landsat satellite images have a 16 day return period and a spatial resolution of 30 m. Landsat images are available from 1972 to the present [41], which makes this product very suitable for land cover classification [42–44] and phenological vegetation monitoring [45,46]. The Landsat images are available from the website of the United States Geological Survey (https://earthexplorer.usgs.gov). The Landsat TM/ETM+/OLI data were originally distributed as a Level 1 Terrain Precision Correction (L1TP) (Path: 120, Row: 34) product. Considering that the product performs the correction process according to the same geometric correction program, L1TP products acquired on different dates on the same path/line can meet the spatial matching accuracy required for pixel-level time series analysis [21]. On 31 May 2003, the Landsat-7 ETM+ Spaceborne Scan Line Corrector (SLC) failed. Therefore, in this study, the "landsat_gapfll.sav" plug-in in the ENVI version 5.3 software (Exelis Visual Information Solutions, Inc., Boulder, CO, USA) was used to compensate for ETM+ data loss [47]. Due to the effects

of atmospheric aerosols, topography, and nearby ground features, the original Landsat TM/ETM+/OLI images contain radiation errors caused by atmospheric absorption and scattering. Therefore, in this study, radiation calibration and the fast line-of-sight atmospheric analysis of spectral hypercubes (FLAASH) module (version 5.3.1; Esri Inc., Redlands, CA, USA) were used to carry out the atmospheric correction of the Landsat images, so that the DN value of the original data was converted to the surface spectral reflectance [22].

The MODIS Terra vegetation index product (MOD13Q1, H/V 27/5) was obtained from the website of the NASA Earth Resources Observation and Science Center (https://ladsweb.modaps.eosdis.nasa.gov/). These data covered the period 01 January to 31 December 2018 with an interval of 16 days and a total of 23-time steps. Given the need to fuse NDVI time-series data with a high spatial and temporal resolution, only the NDVI time-series data of the MOD13Q1 vegetation index product were extracted.

Sentinel-2 NDVI time-series data from 2018 and Landsat NDVI time-series data from 2000–2017 were obtained. The NDVI band was used to establish the phenological patterns of different plants:

$$NDVI = (\rho_{NIR} - \rho_{RED})/(\rho_{NIR} + \rho_{RED}) \tag{1}$$

where $\rho$ represents the reflectance. *NIR* represents the near-infrared band, and *RED* represents the red band.

### 2.4. Identification Method of Apple Orchard Planting Year

This paper proposes a new method for the determination of apple orchard planting year on a regional scale. The first important statement here is that we were tracing back based on the latest orchard distribution monitoring results for 2018 to determine which orchard in each year has remained consistent with 2018 so that we can determine the continuous planting year of the orchard. Therefore, the monitoring method provided in this paper is not to monitor the growing year positively from the planting of orchard seedling but to use time-series remote sensing data to reverse determine the continuous growth age of each existing orchard. This method is implemented in the following two steps (Figure 4):

(i)   Extract the spatial distribution of apple trees. The ESTARFM algorithm is used to fuse the 2018 Sentinel-2 NDVI and MODIS NDVI images to generate a fused NDVI time series. Combined with phenological vegetation features generated using the TIMESAT 3.3 toolbox, an SVM supervised classification method is used to extract the spatial distribution of apple trees. Then, the extracted distribution area of apple trees is used as the region of interest for identifying the planting year;

(ii)  Identification of apple orchard planting year. Based on the ROI data of different vegetation types obtained from the field survey, the NDVI time series were extracted from the Sentinel-2/MODIS NDVI spatiotemporally fused images. From the NDVI time-series curves of the apple orchards, three characteristic phenological periods of apple trees were identified, and the NDVI time series of the apple orchards were used as a template to extract the apple orchard distribution area from 2000 to 2017. Then, combined with the Landsat NDVI time-series images composed of three characteristic phenological periods each year from 2000 to 2017 and the apple orchard phenological curve, the apple tree distribution area was calculated for each year between 2000 and 2017 using the ED method, which was run using the MATLAB R2016a software (MathWorks, Natick, MA, USA). Finally, considering the inter-annual changes in apple tree coverage, a pixel-by-pixel inverse time series calculation program developed using the MATLAB software was used to obtain a distribution map of apple orchard planting year for the study region.

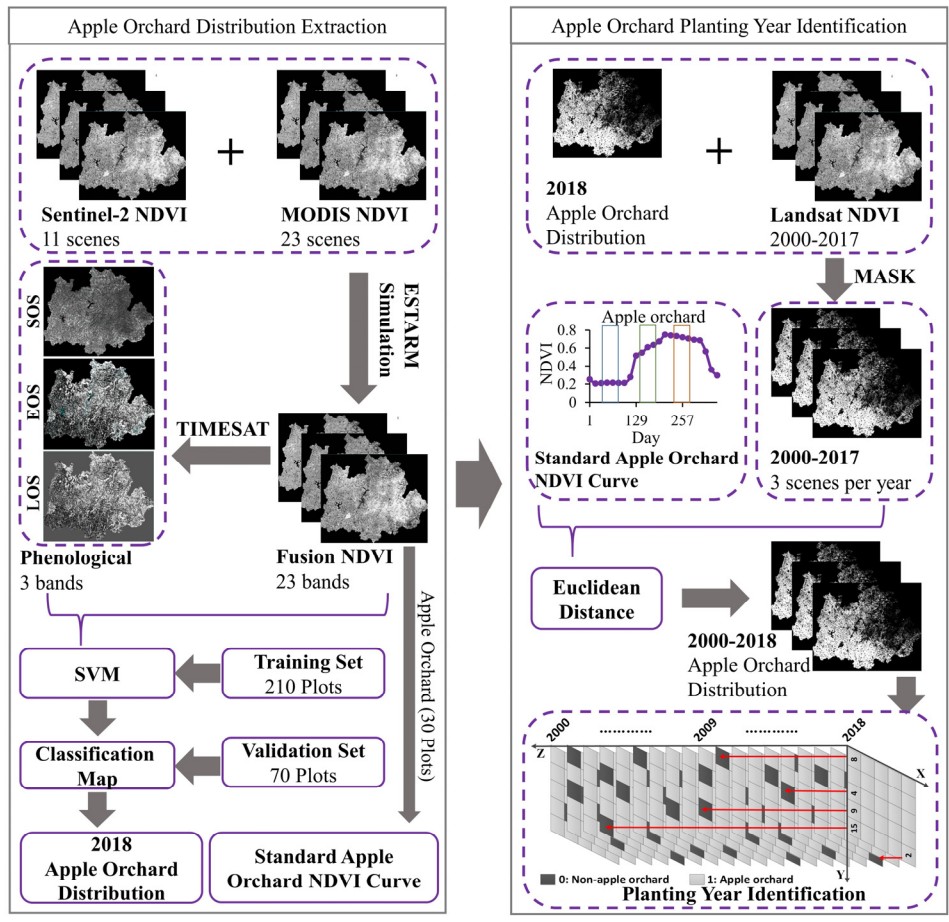

**Figure 4.** Methodology for the extraction of apple orchard planting year and spatial distribution.

### 2.4.1. Phenophase Extraction

In order to extract the phenological characteristics of vegetation in the study area, this study used the ESTARFM algorithm proposed by Zhu et al. to obtain the time series NDVI images [33]. This algorithm introduces conversion coefficients to convert the reflectance changes of the mixed coarse-resolution pixels to the fine-resolution pixels therein so that the reflectance of small objects and linear objects can be accurately predicted. Additionally, the ESTARFM algorithm improves on the original STARFM algorithm by combining the spectral reflectance trends observed at different times and spectral decomposition theory in order to more accurately predict synthetic high-resolution reflection products with heterogeneous landscapes. The ESTARFM algorithm was used to fuse two pairs of Sentinel-2 NDVI (high spatial resolution) and MODIS NDVI (high temporal resolution) images of two-time nodes $(T_i, T_j)$ with MODIS NDVI images at prediction time $T_0$ to generate a fused image with high spatial resolution. In ENVI, a bilinear algorithm is used to resample MODIS NDVI images to the same spatial resolution as Sentinel-2 images (10 m) in order to generate NDVI data similar to Sentinel-2 NDVI data so as to reduce the impact of geolocation errors. Subsequently, the aforementioned band math tools were used to re-scale the Sentinel-2 NDVI data to a pixel value range of 0–10,000, and it was ensured that the raster sizes of the Sentinel-2 NDVI masked MODIS NDVI images were exactly the same. Finally, based on Sentinel-2 and MODIS NDVI data collected from January to December 2018, NDVI time-series data with a spatial resolution of 10 m and a temporal resolution of 16 days were synthesized.

Subsequently, the TIMESAT software was used to extract phenological metrics from the fused NDVI time series in the MATLAB environment. TIMESAT has been widely used in previous studies for the reconstruction of time series vegetation index data and the extraction of phenological information

in various regions and for various vegetation types [23]. Since only one year of Sentinel-2 NDVI satellite data was used and the peak of the vegetation growing season is in the middle of the time series, the TIMESAT toolbox was used to copy the time-series data to generate an artificial time series spanning three years [22]. The artificial time-series data were produced using an iterative and adaptive Savitzky–Golay (S-G) filtering method, which has been proven to be effective for obtaining high-quality time series data from which seasonal parameters can be extracted [48] and for eliminating the effects of noise in NDVI time series data [49,50]. The S-G filter is a filtering method based on local polynomial least-squares fitting in the time domain; it filters out noise while ensuring that the shape and width of the signal are constant, and is widely used for data stream smoothing and denoising. The result of the S-G filtering is that the NDVI time series for each pixel is temporally smoothed, thus eliminating the effects of atmospheric noise and being gap-filled [51]. Then, three phenological indicators were extracted based on the fused NDVI time series, including the beginning of the growing season (BOS), the ending of the growing season (EOS), and the length of the growing season (LOS), respectively. Among them, the extraction of vegetation phenology uses the dynamic threshold method. BOS and EOS are defined as the season start/end time when the fitted NDVI curve reaches an absolute value of 0.4; LOS is defined as the interval between the ending and the beginning of the growing season.

### 2.4.2. Land Cover Classification and Accuracy Verification

According to the distribution range of different vegetation types in the study area, seven land types were selected as the main land cover types, namely apple orchard, cherry orchard, vineyard, coniferous forest, broad-leaved forest, town, and water. In order to improve the accuracy of land cover classification, a classification method combining NDVI time-series data and phenological parameters derived from Sentinel-2 images was developed.

Support vector machine (SVM) classifiers were used to perform the supervised classification of the combined phenological parameters and NDVI time series dataset. The SVM classification method is based on nonlinear kernel transformations of the predictors, projecting them into a higher-dimensional feature space and computing an optimal separation hyperplane [21]. By seeking the minimum structural risk to improve the learning generalization and minimize the risk of experience and confidence, in the case of small sample size, good statistical rules can be obtained. This method has been shown to exhibit excellent performance in most classification cases [22,23,52,53]. In this study, the kernel type of SVM classifier used the radial basis kernel function, and the Gamma in kernel function parameter was set to 0.038, which was the inverse of the number of input image bands. The penalty parameter was then set to 100, which takes into account the balance between sample errors and classification rigidity extensions. Considering that the image data used in this study was the original resolution, that was, Pyramid Levels was set to 0. At the same time, the main land cover type of the study area was included in the training set ROI, so the Classification Probability Threshold was set to 0. Then, using a confusion matrix method, the independent ROI verification sample obtained through the field investigation was used to verify the accuracy of the supervised image classification. Finally, the distribution area of apple trees in 2018 with a spatial resolution of 10 m was obtained and was subsequently used as the area of interest for the determination of apple orchard planting year.

### 2.4.3. Determination and Verification of Apple Orchard Planting Year

Vegetation has regular phenological characteristics, and the NDVI time series usually vary differently for different vegetation types. Additionally, vegetation often has the characteristics of bare soil during the initial planting period. Therefore, in this study, a new method was proposed for identifying the planting year of apple orchards.

First, according to the land cover classification results, the extracted apple orchard distribution area was used as the area of interest. Then, based on the fused NDVI time-series image, a standard apple orchard NDVI phenological curve was extracted for the whole area covered by apple orchards according to the standard curves obtained from 30 apple orchards. The ED [54] equation was used to

extract the distribution area of apple orchards in each year from 2000 to 2017. The equation is given as follows:

$$ED = \sqrt{\sum_{i=1}^{n}(X_i - X_c)^2}\ i \in (1, 2, \ldots, n) \tag{2}$$

where $n$ represents the number of Landsat NDVI bands, $i$ represents a particular band, $c$ represents a particular land cover class, $X_i$ represents the data file values of pixel (x,y) in band $i$, and ED represents the ED from pixel (x,y) to the mean of class $c$.

Landsat satellite images are often disturbed by clouds, which makes it difficult to obtain high-quality satellite images. Therefore, by analyzing the differences of NDVI time series curves among different vegetation types, three characteristic phenological periods that can distinguish different vegetation types are selected. This reduces the number of Landsat images needed to extract apple orchard distribution in 2000–2017 and will improve the computing efficiency of apple orchard extraction. Secondly, using the standard phenological curve of the apple orchard as a template, the similarities between Landsat time-series data consisting of phenological characteristic periods and target type time series data were normalized using the ED method, respectively, to obtain a binarized apple tree distribution map for each year, where the apple tree area is given a value of 1 and the non-apple-tree area is given a value of 0. Finally, using the pixel-by-pixel inverse time series statistical method, binarized images for each year between 2000 and 2018 (i.e., a total of 19 images) were obtained. These images were subsequently used to determine the planting year of apple orchards in the study region.

In this study, the coefficient of determination ($R^2$) was used to measure the correlation between the observed and predicted datasets. Additionally, the root-mean-square error (RMSE) and the normalized RMSE (NRMSE) were used to evaluate the error between the observed and predicted values. Considering that the RMSE has no specific measure for the actual gap between the datasets, the NRMSE was used to compare the difference between the datasets [55]; an NRMSE value below 20% is considered to represent an acceptable simulation accuracy.

$$R^2 = \frac{\sum_{t=1}^{n}(\hat{y}_t - \overline{y})^2}{\sum_{t=1}^{n}(y_t - \overline{y})^2} \tag{3}$$

$$RMSE = \sqrt{\frac{\sum_{t=1}^{n}(\hat{y}_t - y_t)^2}{n}} \tag{4}$$

$$NRMSE = \frac{RMSE}{y_{max} - y_{min}} \tag{5}$$

where $n$ represents the number of samples, $\hat{y}_t$ represents the model-estimated value, $y_t$ represents the measured value, $\overline{y}$ represents the average value, $y_{max}$ represents the maximum value, and $y_{min}$ represents the minimum value.

## 3. Results

### 3.1. Validation of the ESTARFM Method

In order to verify the validity of the ESTARFM method for predicting fused NDVI time-series data, three main phenological seasons were analyzed: spring (25 March 2018), summer (21 June 2018), and autumn (19 October 2018). Then, using the real Sentinel-2 NDVI images for the three seasons and the fused simulated NDVI images for the same seasons, a subregion of the study area of 157 × 153 pixels was randomly selected as the verification region (Figure 5).

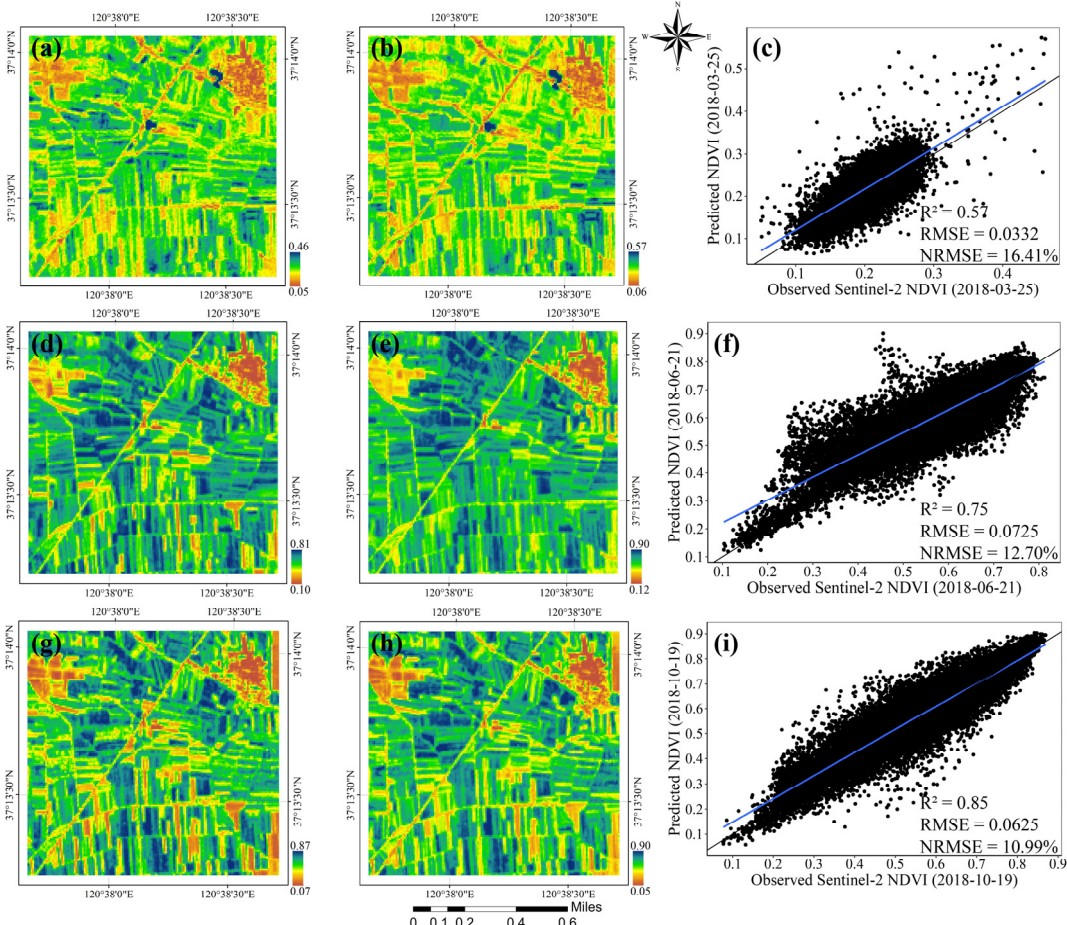

**Figure 5.** Verification of the accuracy of normalized difference vegetation index (NDVI) images fused using the enhanced spatial and temporal adaptive reflectance fusion model (ESTARFM) method. In the figure, the horizontal axis from top to bottom indicates three different seasons: spring (**a**–**c**), summer (**d**–**f**), and autumn (**g**–**i**); The vertical axis from left to right indicates the observed NDVI images (**a**,**d**,**g**), the fused NDVI images (**b**,**e**,**h**), and the verification result (**c**,**f**,**g**).

The results show that, in all three periods, the observed NDVI images and the fused NDVI images have low values of RMSE and NRMSE, which are lower than 0.073 and 16.41%, respectively. This indicates that the fused NDVI images and observed NDVI images in the three periods maintain good consistency. Additionally, the fused NDVI images and the observed NDVI images maintain relatively consistent spatial detail characteristics. However, it was found that the verification accuracy ($R^2$) of the fused NDVI image for spring was 0.57, which is lower than the accuracy in the summer and autumn periods. This may be due to the fact that in spring, the greenness of vegetation changes rapidly, which leads to certain limitations for data fusion in the ESTARFM algorithm.

## 3.2. Extraction of Phenological Information and Analysis

Different vegetation types have different phenological patterns. Therefore, it is very important to understand the changes in the phenological characteristics of different vegetation types before categorizing different land cover types. In this paper, seven vegetative land cover types were selected in the study area, and phenological features were extracted using the TIMESAT toolbox. Using the TIMESAT toolbox, the original NDVI time-series data were first fitted to the S-G function, and phenological features were then extracted. The S-G filter filters out noise in NDVI time-series data while ensuring that the shape and width of the time series data are unchanged. This filter has been proven to effectively eliminate the effects of noise in the NDVI time-series data [49,50].

Figure 6 shows the original NDVI time series curves for the seven land cover types before and after filtering. From the figure, it can be seen that the filtered NDVI time-series data maintains good consistency with the original NDVI data. Compared with the original NDVI time-series data, the filtered curve better shows the seasonal variation trend of different land cover types. Regarding the NDVI time series curves of different land cover types, the NDVI values for urban areas and water are relatively low compared to those of vegetation. Apple orchards, cherry orchards, vineyards, and broad-leaved forests have similar NDVI values, with NDVI values first increasing and then gradually decreasing. However, during the growth period of 145 to 193 days, broad-leaved forests higher NDVI values compared to apple orchards and cherry orchards, and vineyards have the lowest. Furthermore, during the maturity period of 209 to 289 days, the NDVI values of apple orchards are always higher than those of cherry orchards. Coniferous forests maintained a high NDVI value throughout the growth cycle, with only slight fluctuations.

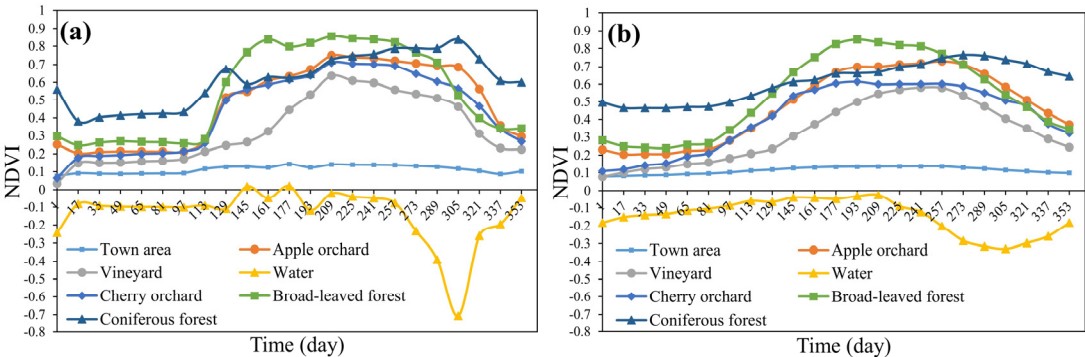

**Figure 6.** Fused NDVI time series data before and after Savitzky-Golay (S-G) filtering. (**a**) Original NDVI time-series curve; (**b**) Filtered NDVI time-series curve.

The three typical phenological features extracted using the TIMESAT toolbox (i.e., BOS, EOS, and LOS), where LOS was shown in Figure 7. Based on the BOS, EOS, and LOS results of the study area, we derived the pixel values of the measured ROI regions corresponding to each vegetation type. Then, we have produced box plots of BOS, EOS, and LOS for five plantations, including the apple orchard, cherry orchard, vineyard, broad-leaved forest, and coniferous forest, to show the differences in phenology among different vegetation (Figure 8). By analyzing the differences in growth and phenological characteristics between different types of vegetation, it was found that the growing season of coniferous forest lasted throughout the year, the growing seasons of apple orchards and broad-leaved forests lasted seven months, and the growing season of vineyards lasted between four and five months. Additionally, it was found that the broad-leaved forest is the first vegetation type to enter the growing season, followed by apple orchards, cherry orchards, and vineyards. Vineyards entered the dormancy period the earliest, followed by the cherry orchard, broad-leaved forest, and apple orchard (coniferous forest does not have a dormancy period).

The results show that different types of vegetation have different growth periods, dormancy periods, and growth cycles. This provides important information for the accurate classification of land cover in the study area.

### 3.3. Classification and Verification

Two different datasets were used to evaluate the impact of phenological features on classification accuracy. One dataset consisted of only the fused NDVI time-series data, and the other was a combination of the fused NDVI time-series data and phenological feature data (as shown in Figure 9). The classification accuracy was evaluated using the confusion matrix calculated based on the verification samples (Table 1).

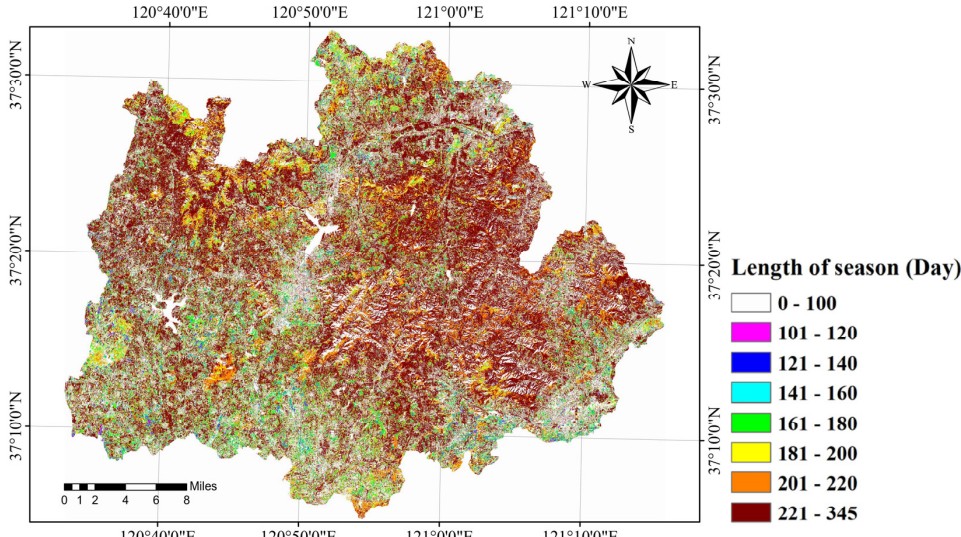

**Figure 7.** Length of growth season—Vegetation growth phenology extracted using the TIMESAT3.3 toolbox.

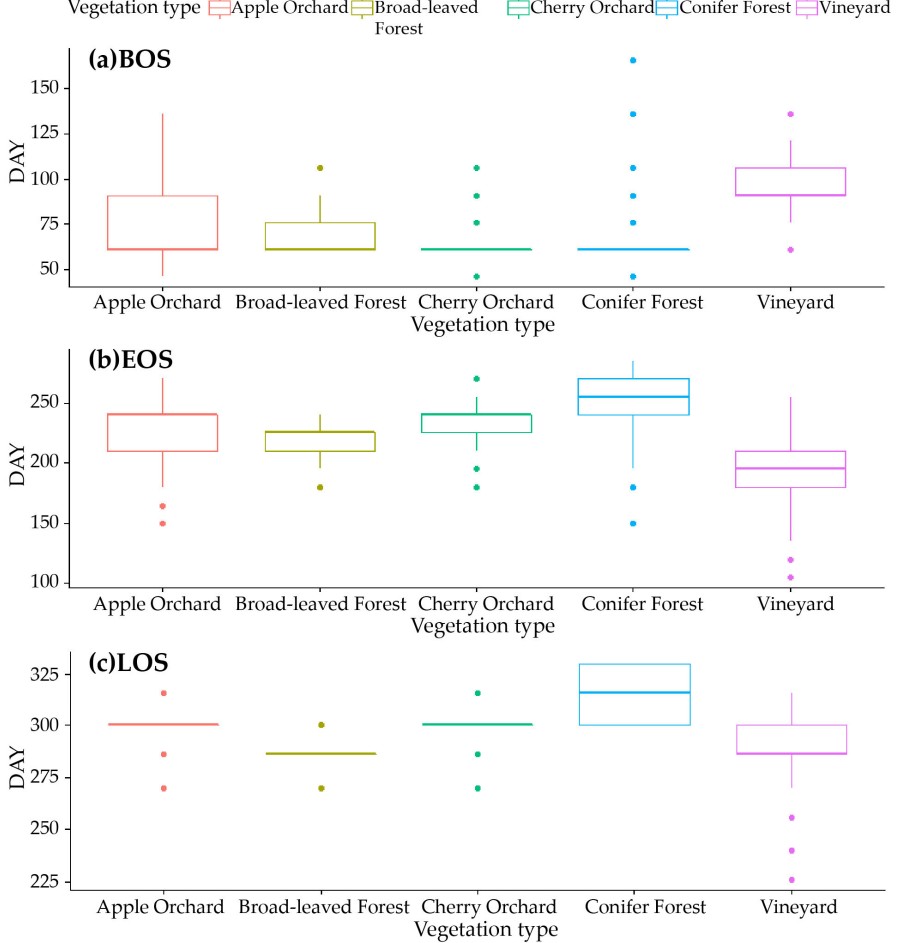

**Figure 8.** The difference in BOS, EOS, and LOS for each vegetation type. Among them, (**a–c**), respectively show the differences of BOS, EOS, and LOS among vegetation.

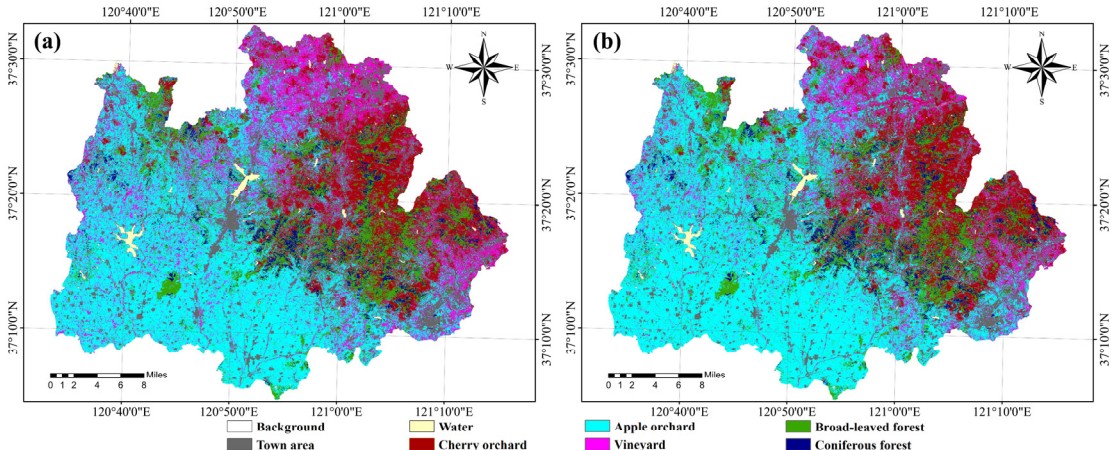

**Figure 9.** Land cover classification map. (**a**) Classification result obtained using only the fused NDVI time-series data; (**b**) Classification results obtained using a combination of the fused NDVI time-series data and phenological features.

**Table 1.** Confusion matrix validation results.

| Class | Fusion NDVI Time Series | | Fusion NDVI Time Series and 3 Phenological Features [1] | |
|---|---|---|---|---|
| | Prod.Acc. (Percent) | User.Acc. (Percent) | Prod.Acc. (Percent) | User.Acc. (Percent) |
| Apple Orchard | 80.17 | 97.30 | 99.66 | 99.15 |
| Vineyard | 89.98 | 93.44 | 98.91 | 95.38 |
| Cherry Orchard | 90.60 | 66.44 | 92.48 | 99.33 |
| Broad-leaved Forest | 100.00 | 97.26 | 100.00 | 98.61 |
| Conifer Forest | 100.00 | 100.00 | 100.00 | 100.00 |
| Town Area | 100.00 | 95.49 | | |
| Water | 97.85 | 100.00 | | |
| Overall Accuracy | 91.7808% (2278/2482) | | 98.1426% (1638/1669) | |
| Kappa Coefficient | 0.9010 | | 0.9751 | |

[1] Three phenological features represent BOS, EOS, and LOS, respectively. Among them, Prod. Acc. represents production accuracy, and User. Acc. represents user accuracy.

The results show that the overall accuracy and Kappa coefficient for the land cover classification which was achieved by combining the 23 fused NDVI time series data and the three phenological feature data were 98.1426% and 0.9751, respectively, while the same parameters for the classification which was achieved using only the fused NDVI time series data were 91.7808% and 0.9010, respectively. Moreover, it was found that in the classification results obtained using only the fused NDVI time-series data, there were more misclassifications among apple orchards, cherry orchards, and vineyards compared to the classification obtained using a combination of the fused data and phenological data, with the user precision for the cherry orchard being 66.44%. This suggests that the use of NDVI time-series data alone cannot effectively distinguish orchard types, while a combination of NDVI time-series data and phenological information can effectively identify different vegetation types, and thus improve the classification accuracy.

Figure 9 shows the estimated regional distribution of the seven vegetation types. It can be seen that apple orchards are mainly distributed in the hilly areas in the southwest of the study area, cherry orchards are mainly distributed in the eastern mountainous areas, and vineyards are distributed in the plain areas in the northeast and southeast. Broad-leaved forests are mainly distributed in high mountainous areas. In terms of areal coverage, apple orchards are the main vegetation type, covering an area of 96,600 hectares (accounting for 48.17% of the total area of Qixia City). The planting areas of cherry orchards and vineyards are 37,880 and 19,326 hectares (18.89% and 9.64% of the total area), respectively. Coniferous forests have a distribution area of 19,913 hectares (9.93% of the total area),

while broad-leaved forests have the smallest distribution area, covering an area of 7733 hectares (3.85% of the total area).

### 3.4. Extraction of Euclidean Distance and Estimation of Apple Orchard Planting Year

The fused NDVI time-series data were cropped based on the actual surveyed apple region as the region of interest, and the apple orchard standard NDVI time series phenological curve extracted from the region of interest was used as a template to extract the apple orchard distribution area from 2000 to 2017. Then, the ED between the apple orchard phenological curve and the measured apple orchard area was calculated on a pixel-by-pixel basis. This study uses the standard NDVI time-series data extracted from the regions of interest (obtained from the 2018 field survey) of the apple orchard as a template. Then, the 2018 NDVI time-series image is cropped according to the area of interest of the apple orchard, and the ED between the apple orchard template and the cropped NDVI time-series image is calculated. Finally, the value of 0.43 corresponding to the median distribution of pixels in the area of interest of the apple orchard is taken as the optimal ED. In order to estimate the planting year of apple orchards, the phenological characteristics of different vegetation types were analyzed, and the phenological characteristics of the apple dormancy period (day 49–97; differentiate coniferous forest and water), growing season (day 129–193; differentiate broad-leaved forests, vineyards, and urban areas), and vigorous growth period (day 225–305; differentiate cherry orchard) were extracted as the standard phenological curve of the apple orchards.

The apple region, which was classified using the combination of fused NDVI time-series data and phenological feature data, was used as the region of interest for the extraction of apple orchard planting year. Then, the Landsat NDVI images of key phenological periods in each year were resampled to a spatial resolution of 10 m, and the Landsat NDVI data was masked using this region of interest to ensure that the fused NDVI images had the same spatial resolution and number of rasters. Then, the ED method was performed to extract apple orchards from Landsat NDVI time series images composed of three key periods each year (Figure 10). The ED was calculated pixel-by-pixel from the template. The closest category is the apple orchard area, whose ED is set to 1; otherwise, the ED is 0.

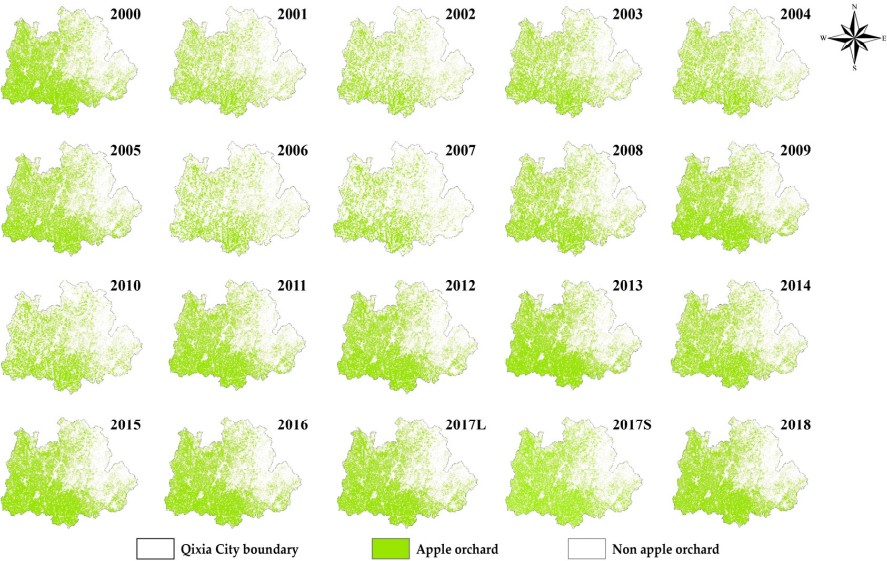

**Figure 10.** Distribution of apple orchards from 2000 to 2018. The distribution for 2000–2017 was extracted using the ED method, and the distribution for 2018 was obtained from the land cover classification results in Section 3.3. Among them, 2017L and 2017S represent the data of Landsat and Sentinel-2 in 2017, respectively.

### 3.5. Apple Orchard Planting Year Extraction and Verification

The binarized apple classification images of the 2000–2018 time series were used to estimate the apple orchard planting year. In this study, the inverse time series pixel-by-pixel calculation method was used to count the apple orchard planting year. Since the satellite images for the initial planting year of the apple orchard show bare soil, these images are classified as non-apple orchard regions (i.e., a pixel value of 0). Therefore, only the entire period of the current orchard from the initial planting year was used as the apple orchard planting year. At the same time, although studies have shown good consistency between the spectral reflectance of Landsat and Sentinel-2, there are still some uncertainties. Therefore, in order to obtain more accurate results, a controlled trial was added to replace the Landsat data of 2017 with the Sentinel-2 data of the same period to calculate the planting years of apple orchards.

The orchard planting year was estimated by calculating the pixel-by-pixel inverse time series accumulation of the binarized apple classification images of the time series, and stop the calculation when a value of 0 is encountered. Then, the planting year calculation of the next pixel is continued until all the pixels of the binarized apple classification images have been traversed. The resulting distribution map of apple orchard planting year is shown in Figure 11. The estimated planting year of a total of 115 apple orchards in the study area was verified using planting-year information obtained from orchard farmers through regional surveys (Figure 12). The verification results show that the identification accuracy ($R^2$) of apple orchard planting year after the replacement of Landsat with Sentinel-2 data in 2017 was 0.747, and the RMSE and NRMSE were 2.56 years and 15.09%, respectively. Compared with the Landsat dataset used in 2000–2017, $R^2$ was improved by 0.019, and the RMSE and NRMSE were reduced by 0.11 years and 0.63%, respectively. This further shows that compared with the Landsat data with a resolution of 30 m, the addition of Sentinel-2 data with a resolution of 10 m can improve the recognition accuracy of the planting year of the apple orchard.

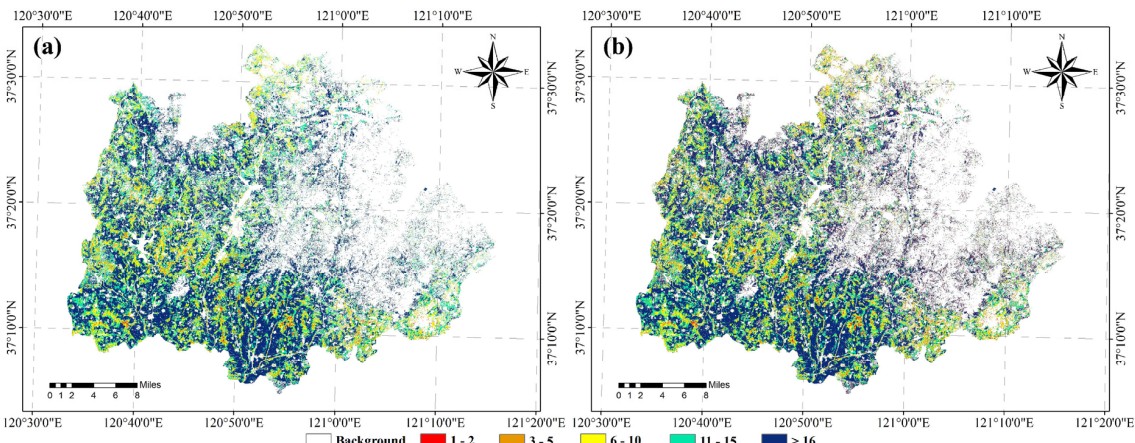

**Figure 11.** Map showing the estimated planting year of apple orchards. (**a**) Recognition results obtained using only the Landsat data from 2000–2017; (**b**) Recognition results obtained by replacing Sentinel-2 data during the same period with 2017 Landsat data.

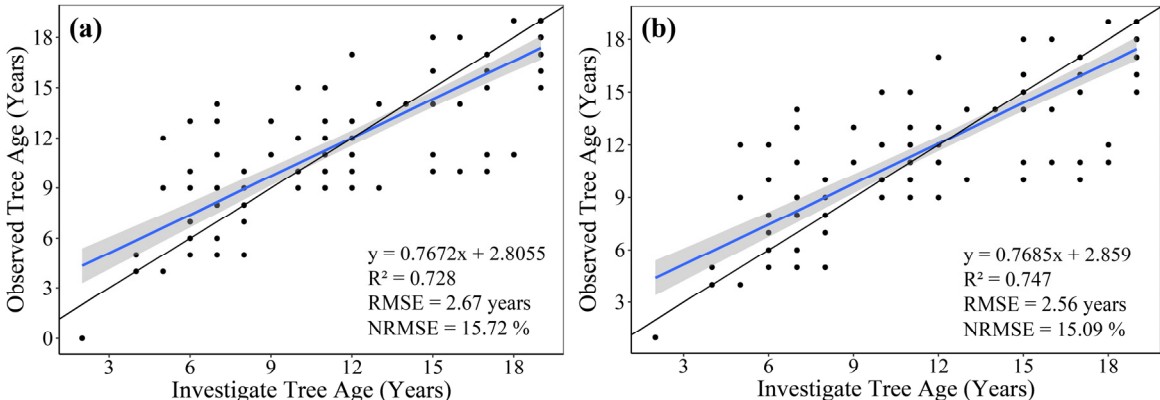

**Figure 12.** Accuracy verification of the estimated planting year of apple orchards. (**a**) Verification results obtained using only the Landsat data from 2000–2017; (**b**) Verification results obtained by replacing Sentinel-2 data during the same period with 2017 Landsat data. The black line represents 1:1 line, and the blue line represents the fitting line between the investigated planting years and the observed planting years.

## 4. Discussion

### 4.1. Limitations of the ESTARFM Method

Significant progress has been made in the use of remote sensing data to predict vegetation phenology [42]. The combination of traditional land cover classification methods and vegetation phenological characteristics can greatly improve the accuracy of land cover classification [22,23]. However, most phenological calculation methods require remote sensing time series data with a high temporal resolution [56]. In this study, Sentinel-2 NDVI data for spring, summer, and autumn were used to verify the accuracy of the fused NDVI data (Figure 5). The results show that the predicted NDVI in summer and autumn has good consistency with the Sentinel-2 NDVI images, while the accuracy of the predicted NDVI is lower in spring.

As spring is the main period to release the dormancy mechanism of vegetation, and the release of the dormancy period is affected by vegetation type, species, or growth environment, which leads to the difference of vegetation growth time [57]. At the same time, during the vegetation growing season, remote sensing images are often disturbed by cloud cover, which makes it difficult to obtain high-quality satellite images with medium or high spatial resolution [58]. Considering the simulation mechanism of the ESTARFM algorithm, this algorithm cannot simulate such sudden changes, resulting in some differences between the fused NDVI images and the actual NDVI images in the corresponding period. This can be attributed to the limitation of the ESTARFM method when the greenness of vegetation changes rapidly (i.e., as occurs in spring), as was noted by Zhu et al. [33]. However, the reduced accuracy of NDVI prediction in spring did not disturb the shape of the entire vegetation phenology curve. The S-G smoothing algorithm is used to smooth the fused NDVI time series, which can better maintain the shape of the vegetation time series phenology curve [59].

In addition, considering the small orchard scale and broken distribution in the study area, this study used summer Sentinel-2 NDVI observation images and corresponding NDVI fusion images to verify the possible impact of this situation on image fusion. In this paper, two different orchard distribution areas were synchronously selected as controls (uniform and heterogeneous orchard distribution areas; plot size: $100 \times 100$ pixels). The results in Figure 13 showed that the accuracy of the NDVI image of the uniformly distributed orchard is better than the unevenly distributed orchard area, the $R^2$ reached 0.74, and the accuracy was relatively improved by 0.25; RMSE and NRMSE reached 0.0836 and 13.76%, and the errors were reduced by 0.0048 and 2.56%, respectively. This showed that the simulation accuracy of the ESTARFM algorithm is affected by the spatial heterogeneity of the study area, and the better the spatial uniformity of the study area, the higher the simulation accuracy. At

the same time, the lower-precision fused images will interfere with the accuracy of the apple orchard distribution extraction and further affect the accuracy of the orchard planting year. Therefore, the ESTARFM algorithm needs to be further improved to strengthen the applicability of the algorithm in a variety of spatial distribution scenarios in the future.

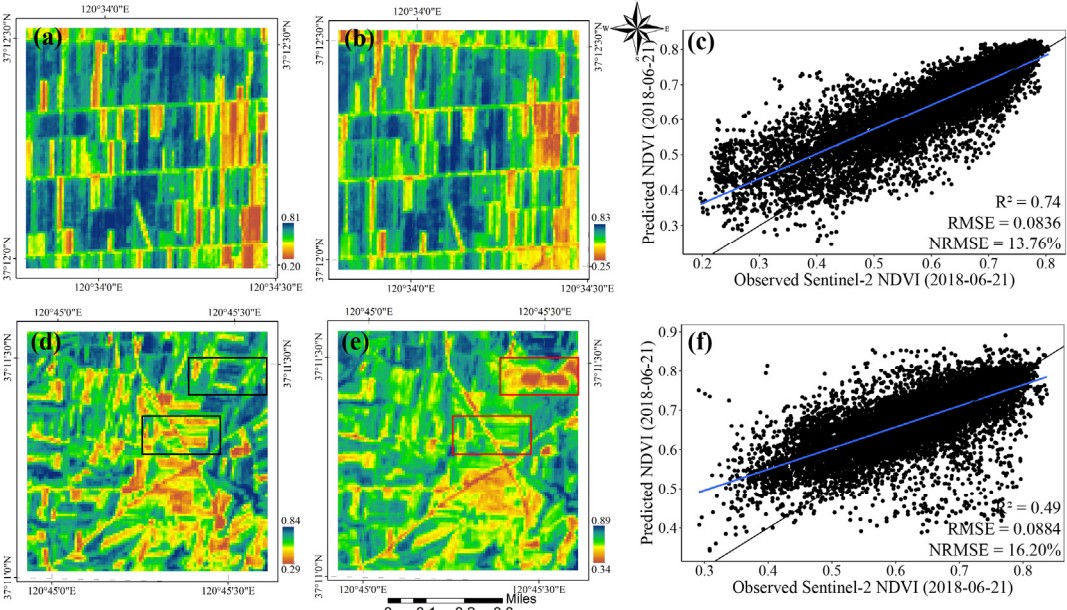

**Figure 13.** Differences in spatial heterogeneity of orchard distribution on the accuracy of fusion NDVI images. In the figure, the horizontal axis from top to bottom indicates two different plots: homogeneous plot (**a**) and (**b**), plot with spatial heterogeneity (**d**) and (**e**); The vertical axis from left to right indicates the observed NDVI images (**a**) and (**d**), the fused NDVI images (**b**) and (**e**), and the verification result (**c**) and (**f**); The black-framed area in (**d**) represents the real NDVI image, and the red-framed area in (**e**) represents the abnormal NDVI fusion image.

*4.2. Method for the Determination of Apple Orchard Planting Year*

Most previous research on the remote sensing identification of vegetation age focused on spectral characteristics [8–13], textural characteristics [14], or forest structure parameters [15] during the vigorous growth period. However, this places higher requirements on the spatial resolution of remote sensing images and the density of lidar point clouds, which limits the size of the monitoring area. Additionally, since different orchards may employ different management and cultivation methods, trees may have similar outlines but different ages, which makes it difficult to judge the actual planting age based on the spectral, textural, or structural characteristics. Therefore, the monitoring method provided in this paper is not to monitor the growing year positively from the planting of orchard seedlings but to use time-series remote sensing data to reverse determine the continuous growth age of each existing orchard. Compared with traditional remote sensing-based methods for the estimation of vegetation age, the method proposed in this work does not need to consider the spectral texture or structural characteristics of plants; rather, it comprehensively considers the phenological characteristics of vegetation, thus greatly improving the classification accuracy, and has a wider monitoring range and stronger generalization ability.

However, considering the 16-day long regression period of Landsat satellites and the data requirements of about 20 years in this study, it is difficult to obtain high-quality cloudless satellite images, which in turn affects the accuracy of apple orchard distribution extraction in this area. In addition, compared with the NDVI time series curve consisting of 11 cloudless Sentinel-2 observational images, the curve composed of the fused NDVI time series data is more representative of the entire

growth period of the vegetation (Figure 14). However, the NDVI time-series curves of the apple trees and cherry trees extracted in the two datasets tend to be similar. The phenological characteristics of the apple orchards and cherry orchards are most similar in their dormancy and growing seasons, with only minor differences during the vigorous growth period. In the Sentinel-2 NDVI time series data after S-G filtering, there was no significant difference between the apple trees and the cherry trees. However, in the fused NDVI time-series data after S-G filtering, in the vigorous growth period, the apple trees and cherry trees showed obvious differences. At the same time, there may be some phenological differences among different vegetation species. The extraction of apple orchard distribution based on phenological characteristics alone may have some misclassifications, which may affect the accuracy of planting year extraction.

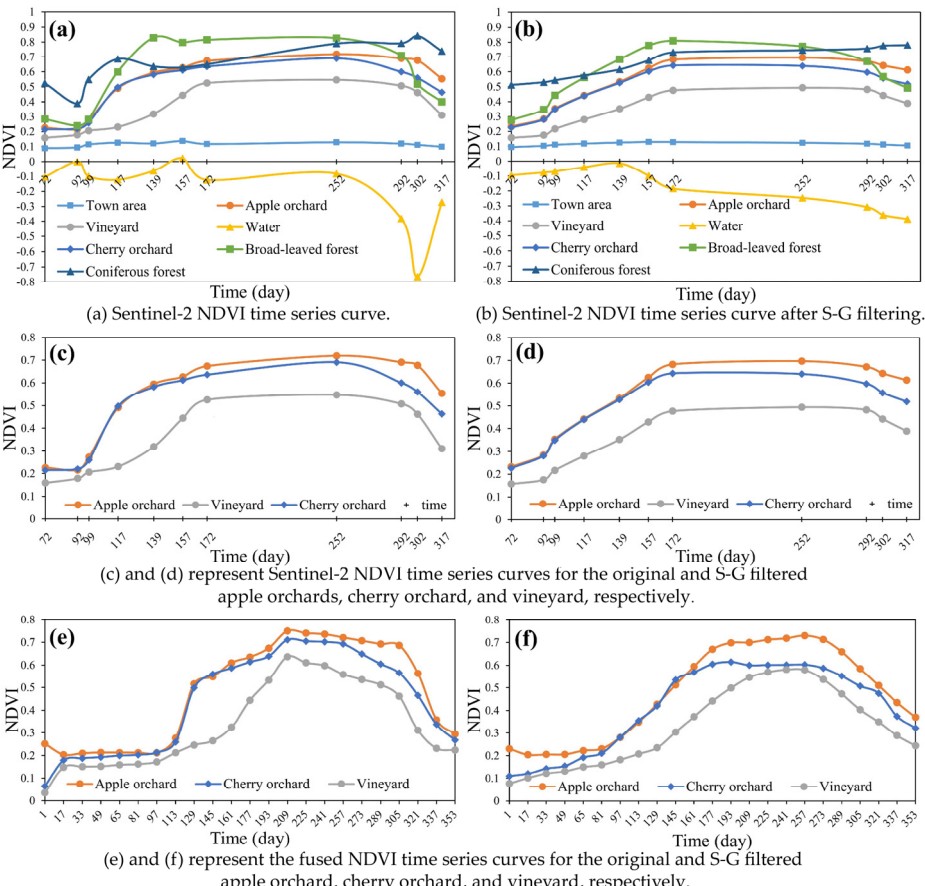

**Figure 14.** Differences in the NDVI phenology curves based on cloudless Sentinel-2 NDVI data (11 images) and those based on fused NDVI time-series data (23 images).

### 4.3. Application Prospect of Apple Orchard Planting Year

Information about apple orchard planting year is essential for sustainability assessments, yield forecasting, and precision orchards. At the same time, this information can assist the government or orchard management department in constructing orchard policies or decisions to support the further development of the orchard. Previous studies have shown that, given the large scale of some plantations, plantations with different tree ages should use specific water, fertilizer, and pesticides schemes, which is essential to effectively improve resource utilization and productivity per unit area [14,60]. In addition, due to the general decline of productivity caused by the aging of the orchard, it is necessary to obtain the planting year information of apple orchard at the regional scale in time, and assist the local government in formulating the aging transformation policy of orchard. With the increase of the planting year of apple orchards, there is often a certain degree of breeding of pests and

diseases. Therefore, the identification of planting year can provide guidance for the prevention and control of diseases and pests in large-scale orchard. At the same time, information about apple orchard planting year is also very helpful to the production forecast, which is conducive to the market planning and operation before and after the apple harvest, which is crucial to the sustainable development of the local economy. This information is valuable for the government agencies to use as a decision-making tool for various agronomic management needs, such as adjusting the agricultural input, forecasting the harvest, and allocating funds for new replanting schemes.

## 5. Conclusions

Therefore, the monitoring method provided in this paper is not to monitor the growing year positively from the planting of orchard seedlings but to use time-series remote sensing data to reverse determine the continuous growing age of each existing orchard. This method is based on a combination of (1) a fusion of two kinds of remote sensing data with different temporal and spatial resolutions and (2) phenological information. The verification results show that the identification accuracy ($R^2$) of apple orchard planting year is 0.747, and the RMSE and NRMSE are 2.56 years and 15.09%, respectively. In addition, the fused images for the three growing seasons of spring, summer, and autumn were verified using Sentinel-2 images from the same period. The results showed that the ESTARFM algorithm could not effectively capture the rapid changes in vegetation which occur in spring, and the simulation accuracy ($R^2$) of the image is only 0.57, and the RMSE and NRMSE are 0.0332 years and 16.41%, respectively. However, S-G filtering can effectively eliminate the variation of NDVI during this period. Then, two different datasets were used to evaluate the impact of phenological features on classification accuracy. The results show that the combination of the NDVI time-series data and phenological information can effectively improve the classification accuracy (the overall accuracy and kappa coefficient are 98.1467% and 0.9775, respectively). Subsequently, using the extracted apple orchard NDVI time-series data, the distribution of apple trees in the study area between the years 2000 and 2017 was calculated using the ED method. Based on the calculated distribution of apple trees, the planting year distribution of apple orchards in the study region was obtained by using a pixel-by-pixel inverse time series calculation method. Finally, the accuracy of the estimated planting year distribution of apple orchards was verified using farming records obtained from regional surveys. The method for the determination of apple orchard planting year proposed in this study can be applied to other types of orchard or plantations to assess the fruit yield and the stocking volume of plantations in various areas.

**Author Contributions:** Conceptualization, G.Y.; data curation, Y.Z., H.Y. and J.W.; investigation, Y.Z. and H.Y.; methodology, Y.Z., G.Y. and H.Y.; project administration, G.Y.; resources, G.Y.; software, Y.Z., J.W., L.L. and L.F.; supervision, G.Y.; validation, Y.Z., J.W. and F.Z.; visualization, Y.Z.; writing—original draft, Y.Z.; writing—review and editing, G.Y. and C.Z. All authors have read and agreed to the published version of the manuscript.

**Funding:** This study was supported by the National Key Research and Development Program of China (2017YFE0122500 and 2016YFD0700303), the Beijing Natural Science Foundation (6182011), and the Beijing Academy of Agriculture and Forestry Sciences (KJCX20170423).

**Conflicts of Interest:** The authors declare no conflict of interest.

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
