# Peer review of "Identification of Apple Orchard Planting Year Based on Spatiotemporally Fused Satellite Images and Clustering Analysis of Foliage Phenophase"

_remotesensing, doi:10.3390/rs12071199_

Round 1

Reviewer 1 Report

The manuscript titled “Identification of apple orchard planting year based on spatiotemporally fused satellite images and clustering analysis of foliage phenophase” (Manuscript ID: remotesensing-740916) presents a new approach for the age identification of apple orchards using multi temporal satellite imageries. My main concerns about the manuscript are:

  1. The authors stated that they used the default parameters in SVM classification (Page 9, Line 282). However, these parameters can have huge impacts on the classification output and default values were rarely optimal. Therefore, the usage of a parameter selection method such as cross-validation or more advanced method is essential.
  2. Please correct the Figure 1. (c) Caption as “Location of the samples on the field”.
  3. In Page 5, Line 163 the authors stated that “The Sentinel-2 and Landsat data are shown in Figure 3”. However in figure 3 we don’t see the images, Figure 3 shows the availability of the images.
  4. In the manuscript, the apple orchards distributions between the years 2000 to 2017 were extracted. However, in Page6, Line 201 it was stated that Landsat NDVI time series for 1999-2017 were obtained? Why 1999?

Reviewer 2 Report

Examples of English and grammar review required can be found on numerous lines.  For example, line 275, the "&" is used rather than the word "and" - please avoid use of symbols and spell out words.

There are quite a few issues with agreement between plural and singular throughout.

What I liked about this paper was the phenological clustering, but please explain the 3 key metrics in more detail - the BOS, EOS, and LOS.

I also liked the NDVI timeseries analysis, but I was not totally clear how this is converted into the age of the tree.  Please explain better.

I have seen many papers on ESTARFM, and I realize here it is just a fusion method.

Reviewer 3 Report

orally fused satellite images and clustering analysis of foliage phenophase  

Yaohui et al. 2020 

A brief summary (one short paragraph) outlining the aim of the paper and its main contributions. 

The study “Identification of apple orchard planting year based on spatiotemporally fused satellite images and clustering analysis of foliage phenophase” by Yaohui et al. 2020 proposes a method for identifying the year of planting of apple orchards by using a remote sensing approach. To that purpose a time-series of optical remote sensing data made of Sentine-2 (for 2019), Landsat (2000-2017) is being used. The ESTARFM fusion algorithm is being used in creating cloud free Sentine-2 NDVI time-series. The paper employs an interesting approach in a field highly important today (precision farming), an proposes innovative methods. In spite of that, some elements of the research design are questionable and contribute to uncertainty – they should be replaced or better explained.   

Broad comments  

Regarding the landcover classification some inconsistencies have been found.  

Firstly, phenological parameters are specific for vegetation. Therefore, calculating such parameters for water and artificial surfaces (town area) makes no sense. In my opinion, a more suitable approach regarding the land cover classification would be to do it in two steps:  

1) First classification using just fusion NDVI time-series for all land cover classes (including water and town area). The results for water and town should be considered final in this step.  

2) The study area should be masked for water and town area with the result from the first step. A second classification using the fusion NDVI time-series + 3 phenological features should be computed just for the vegetation classes.  

A second concern I have regarding the land cover classification, is that the classes chosen might not capture the variety of vegetation that can be found in your study area. More precisely, you don’t have a class for grassland and/or bare soil. At a spatial resolution of 10m, it would be important to have the bare soil area/grassland area in your reference year (2018) – so as not to have it confused with orchards. 

Regarding the use of Landsat for the period 2000 – 2017.  

As far as I know, Sentinel-2 data is available 2016 onwards. For more accurate results, I think for 2016 and 2017, Sentinel-2 data should have been used. There are a lot of papers which outline the good agreement between Landsat and Sentinel-2 spectral reflectance, however, some uncertainty is still there and it would have been interesting to compare the Sentinel-2 results between the years with available data.  

Regarding the use of euclidian distance to indentify time-series of apple orchards in Landsat data. 

Is there any literature suggesting that using the Euclidian distance for identifying similar vegetation time-series has shown good results?  Time series of vegetation (in your case apple orchards) in extreme years with drought or a prolonged cold/hot period could show significantly different from your reference time-series from 2018. How was the optimal (median) Euclidian distance been chosen?  

Specific comments  

In chapter 3.1 Phenophase Extraction – an ilustration would have been interesting to see – regarding the difference in BOS, EOS and LOS for each vegetation type.  

Rows 302 – 305 – Phrase not clear. 

Rows 282-286 & Table 1 – is the confusion matrix computed using the validation data? Or is it done using still the training data? Same question regarding the Kappa Coefficient. 

Row 420: how was the median determined as the optimal threshold? 

Figure 9: there should be a different color between no data (outside study area borders) and no orchards. 

Chapter 2.2 Field data – how were the DOYS of the phenological stages established? Through literature? Is there no variation between the years (years with extreme weather)?  

Figure 3: Does the number in each row stand for the day of year of the respective phenological stage? I.e. 68-7 in the dormant period (first row) - stands for day 68? How was this measured? 

Row 167: link to scihub needs to be corrected (without /dhus/#/home) 

Figure 11: Regression lines – explain which line represents which datasat I.e. blue line – estimate planting years, black line - observed planting years 

Figure 13 & general for all figures – better if it would be written under every graph what it depicts.  

Round 2

Reviewer 1 Report

The authors revised the paper according to my comments. 

Author Response

Thank you very much for your comments.

Reviewer 3 Report

Review of the revised version of ” Identification of apple orchard planting year based on spatiotemporally fused satellite images and clustering analysis of foliage phenophase”

Broad comments for the author:
Response to replies to comment 1
I agree that setting a threshold of 0.4 of the seasonal amplitude will exclude time-series of water bodies and settlements – as they have characteristically very low NDVI values. If they did not participate in the phenological calculation, which 3 (phenological) features were used in running the second SVM features in addition to the fusion NDVI time-series (as shown in the confusion matrix for Table 1)?
I agree that there is no significant change in the confusion matrix – but calculating phenological characteristics especially for artificial areas (town area) is scientifically wrong. This pops up immediately into ones mind when looking at Table 1.
I think you should at least take out the two classes (town area and water bodies) out of the second part of Table 1 “Fusion NDVI time series and 3 phenological features”.

Response to replies to comment 3
Thank you for your response. I totally agree that a better resolution can improve the results.
A question regarding Figure 10: for b) did you mean replacing Landsat 2017 with Sentinel-2 2017? Otherwise if a) is made by using just LANDSAT I don’t get what you did exactly.

Satisfactorily addressed:
Comment 2
Comment 4 Questions 1 & 2

Specific comments for the author:
Response to replies to comment 1:
Yes, the figure that you provided is very good for visualizing the difference between the 3 phenological features. I would include it to the manuscript because I feel it adds value to it.
All comments have been satisfactorily addressed.
